# Positive selection underlies repeated knockout of ORF8 in SARS-CoV-2 evolution

Cassia Wagner [1,2] ✉, Kathryn E. Kistler [2,3], Garrett A. Perchetti [4], Noah Baker [4], Lauren A. Frisbie [5], Laura Marcela Torres [5], Frank Aragona[5], Cory Yun[5], Marlin Figgins [2,6], Alexander L. Greninger [2,4], Alex Cox[5], Hanna N. Oltean[5], Pavitra Roychoudhury [2,4] & Trevor Bedford[2,3]

Knockout of the ORF8 protein has repeatedly spread through the global viral population during SARS-CoV-2 evolution. Here we use both regional and global pathogen sequencing to explore the selection pressures underlying its loss. In Washington State, we identified transmission clusters with ORF8 knockout throughout SARS-CoV-2 evolution, not just on novel, high fitness viral backbones. Indeed, ORF8 is truncated more frequently and knockouts circulate for longer than for any other gene. Using a global phylogeny, we find evidence of positive selection to explain this phenomenon: nonsense mutations resulting in shortened protein products occur more frequently and are associated with faster clade growth rates than synonymous mutations in ORF8. Loss of ORF8 is also associated with reduced clinical severity, highlighting the diverse clinical impacts of SARS-CoV-2 evolution.

Selection pressure on SARS-CoV-2 has shaped the population of circulating virus since its emergence in humans. The virus has undergone repeated selective sweeps of variant of concern viruses, such as Delta and Omicron, and more recently by lineages within-Omicron, including BA.2 and XBB, in which increased fitness derives from mutations contributing to both intrinsic transmissibility and immune escape[1–11]. Adaptive mutations are overrepresented in spike, the viral entry protein and primary target of protective adaptive immunity, and mutations here alter tropism, improve transmission, and evade host immunity[12–17]. The number of mutations in S1, the spike subunit containing the receptor binding domain, correlate with viral growth rate[18].

Adaptive evolution has not been limited to spike, however. Specific missense mutations in open reading frames (ORFs) for nonstructural (ORF1a and ORF1b), other structural (nucleocapsid, N) and accessory (ORF3a) proteins are also associated with increased viral fitness[19,20]. ORF8 has repeatedly been knocked out during SARS-CoV-2 evolution, though the evolutionary pressures acting on loss of ORF8 are not known. Multiple large deletions of ORF8 and occasionally neighboring ORF7a and ORF7b have been identified around the world,

including in Singapore in 2020, where it was associated with reduced clinical severity[21–25]. Additionally, premature stops in ORF8 causing early truncation of the 121-amino acid protein have been reported, including in mink and pangolin animal species, the Alpha variant of concern (Q27*) and lineage XBB.1 descendants (G8*)[3,11,26–28]. As of September 2023, the vast majority (~90%) of currently circulating SARS-CoV-2 has ORF8 knocked out[29]. This pattern mirrors SARS-CoV's loss of ORF8 after introduction into humans[30].

ORF8 is a viral accessory protein that aids in immune evasion[31]. As a secreted protein, it drives an early antibody response[32,33], potentially acting as a decoy for protective adaptive immunity. Many functions have been attributed to ORF8, including downregulating major histocompatibility complex class I (MHC I)[33–35], decreasing antibody dependent cellular cytotoxicity activity[36], inhibiting Type I IFN production[37–40], suppressing IFN-γ induced antiviral gene expression[41], and disrupting host epigenetic regulation by acting as histone H3 mimic[42]. In its unconventional, unglycosylated state, ORF8 may contribute to cytokine storms by activating the IL-17 pathway[43–45].

[1]Department of Genome Sciences, University of Washington, Seattle, WA, USA. [2]Vaccine and Infectious Disease Division, Fred Hutchinson Cancer Center, Seattle, WA, USA. [3]Howard Hughes Medical Institute, Seattle, WA, USA. [4]Department of Laboratory Medicine and Pathology, University of Washington, Seattle, WA, USA. [5]Washington State Department of Health, Shoreline, WA, USA. [6]Department of Applied Mathematics, University of Washington, Seattle, WA, USA. ✉e-mail: cassiasw@uw.edu

Given these varied potential functions of ORF8, its repeated knockout is perplexing. One hypothesis is that ORF8 knockout is deleterious to SARS-CoV-2 fitness but rose to fixation frequency by hitchhiking along with fitness enhancing mutations in Alpha and again in XBB.1 descendants. Another hypothesis is that ORF8 knockout has no impact on viral fitness, and the gene is undergoing neutral evolution. Here, again, fixation could be explained by hitchhiking. A final hypothesis is that ORF8 knockout improves viral fitness, and positive selection for knockout has contributed to its global spread.

To explore these hypotheses, we use SARS-CoV-2 sequences from Washington State (WA) from February 2020-March 2023 to determine prevalence of ORF8 knockout across time, contrasting this with the knockout of other SARS-CoV-2 genes. Here, we can observe knockouts occurring on a variety of fitness backgrounds, not just the fit viral backbones which swept globally. Next, we use a large, global phylogeny of SARS-CoV-2 to compare expected counts and clade growth rates of nonsense mutations, which truncate the ORF8 protein, to synonymous mutations in ORF8. Finally, we assess linked hospitalization and death data to determine the clinical impact of ORF8 knockout.

## Results

### Prevalence of ORF8 knockout in WA State

We quantified how often ORF8 was knocked out during SARS-CoV-2 evolution in WA from the beginning of the COVID-19 pandemic through March 2023. Our dataset included knockouts under a wide potential array of selection pressures, including knockouts which primarily spread locally and knockouts in the Alpha and XBB.1 descendant

viruses which spread globally. As the first U.S. state to detect community transmission of SARS-CoV-2, WA has robustly sequenced COVID-19 cases throughout the pandemic aided by a sentinel surveillance sequencing system for geographic coverage[46–48]. From April 2021 through March 2023, 17.25% of all COVID-19 infections in WA were sequenced, with the lowest sequencing coverage in December 2021 (3% of cases) and the highest in February 2022 (28% of cases). This high sequencing coverage makes WA an ideal location to understand prevalence of ORF8 knockout across time[49].

We considered samples to contain a potential knockout in ORF8 if they contained a large deletion (>30 bp) or a premature stop codon resulting in at least a 10 codon shorter protein coding sequence. This cutoff, though arbitrary, prevents mislabelling common, short deletions as knockouts while avoiding preferentially maximizing or minimizing knockouts in any one gene (Supplementary Fig. S1). Samples with a mutation known to cause amplicon dropout in ORF8 were excluded from potential knockouts (see methods). We identified 14,929 samples with a potential knockout of ORF8, representing 11.7% of high coverage (≥95%) SARS-CoV-2 sequences collected in WA through March 2023 (Fig. 1A). For ORF8, the number of knockouts was robust to cutoff length: with a cutoff of 95 codons missing, 9.9% of sequences would still have an ORF8 knockout (Supplementary Fig. S1).

While the majority of ORF8 knockouts were found in variants descending from clade 20I (Alpha), clade 22F (lineage XBB), clade 23A (lineage XBB.1.5), and clade 23B (lineage XBB.1.16), ORF8 knockout also occurred in an average of 3% of all other clades (Fig. 1B). Most knockouts were due to premature stop codons, either from nonsense

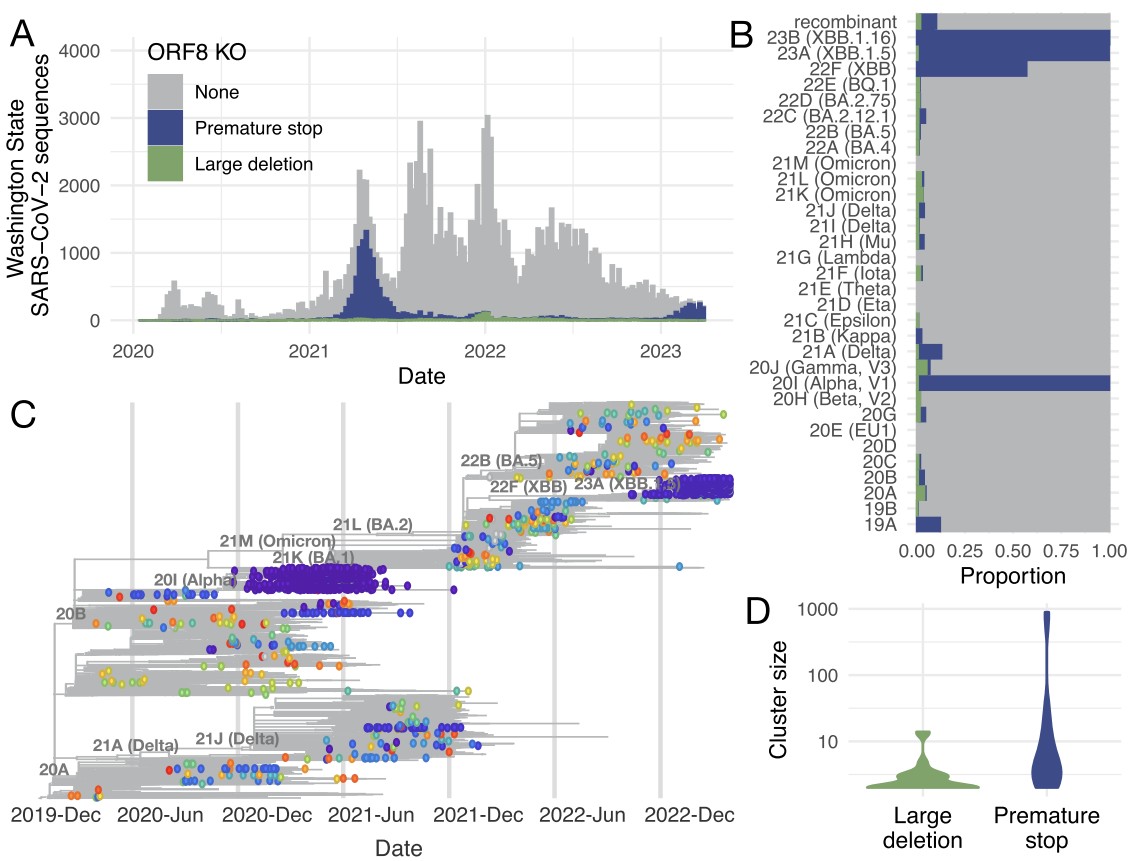

**Fig. 1 | ORF8 is repeatedly knocked out during SARS-CoV-2 evolution in Washington State. A** Distribution of the number of SARS-CoV-2 sequences collected in WA by collection date. Histogram is colored by the type of potential ORF8 knockout (none = gray, premature stop = blue, large deletion = green). **B** Proportion of sequences with a potential ORF8 knockout by Nextstrain Clade. **C** Time-resolved phylogenetic tree of 16,268 SARS-CoV-2 sequences enriched for

sequences in WA (9,854) evenly sampled across time through March 2023. Tips with a potential ORF8 knockout are shown as circles colored by a unique cluster. There are 355 unique clusters, so colors are reused, but adjacent tips of the same color belong to the same cluster. All other tips are plotted as gray lines. **D** Violin plots of cluster size for ORF8 knockouts due to large deletions (green) or premature stops (blue). Source data are provided as a Source Data file.

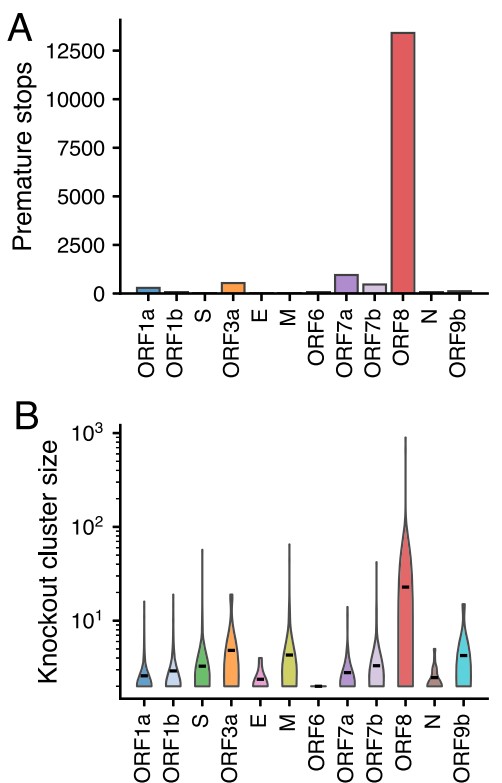

**Fig. 2 | ORF8 has more premature stops and larger knockout clusters than any other gene. A** Number of premature stops by gene in WA SARS-CoV-2 sequences through March 2023. **B** Size of parsimony clusters with a gene knockout due to large deletion or premature stop for all SARS-CoV-2 genes. Clusters were reconstructed from the maximum likelihood phylogenetic tree enriched for WA sequences with even temporal sampling. Source data are provided as a Source Data file.

or frameshift mutations, with only 10.2% of knockouts being large deletions. This suggests that most knockouts are real and not artifactual errors in sequencing as point mutations and small gaps can be confidently inferred with short read sequencing and reference-based genome assembly.

We constructed a phylogenetic tree enriched for sequences sampled in WA spread evenly across time to determine if potential knockouts clustered together phylogenetically. We identified parsimony clusters of ORF8 knockouts across the tree using unidirectional clustering for large deletions and bidirectional clustering for premature stops (see methods) (Fig. 1C). We identified 355 unique clusters: 250 large deletion clusters and 105 premature stop clusters. Most clusters were singletons, with only 53 clusters containing at least two samples. Premature stop clusters were larger with a mean cluster size of 17.2 compared to 1.2 for large deletions ($p = 2.3e{-}04$, Wilcoxon Rank Sum test, two-sided) (Fig. 1D). This difference in cluster size could reflect different fitnesses associated with different types of gene knockout. For example, 7 out of 27 non-singleton, large deletion clusters resulted in knockout of ORF8 and early truncation of ORF7b, which could result in altered fitness compared to ORF8 knockout alone. However we did not observe a difference in size between deletion clusters only knocking out ORF8 and deletion clusters also affecting ORF7b ($p = 0.14$, Wilcoxon Rank Sum test, two-sided). Among non-singleton clusters, deletion size was positively correlated with cluster size (Pearson's $r = 0.47$, $p = 0.012$), and this effect was robust to excluding deletions that also truncated ORF7b (Pearson's $r = 0.48$, $p = 0.028$) (Supplementary Fig. S2A). Rather than resulting from a difference in fitness, it is more likely the difference in cluster size by knockout type occurs because many potential large deletions

represent a sequencing error, rather than a large deletion, and fail to cluster with other potential large deletions.

To determine whether potential large deletions were true deletions versus amplicon dropouts or sequencing errors, we screened a subset using PCR and Sanger sequencing. Of 9998 University of Washington samples available at the time of screening, 120 were found to have sequences with contiguous strings of N's (>266 bp) from ORF7a through ORF8. Of these, 89 samples had sufficient volume and quality for PCR and Sanger sequencing, and 23/89 (25.8%) were confirmed to have large deletions (≥344 bp) (Supplementary Table S1).

For knockouts with premature stop codons, we did not find a correlation between truncated protein length and cluster size (Pearson's $r = -0.14$, $p = 0.49$) (Supplementary Fig. S2B). However, 77% of non-singleton knockout clusters due to an early stop mutation were predicted to have a truncated protein of 26 codons or smaller (Supplementary Fig. S2D). This skewed distribution suggests that most premature stops are causing gene knockouts rather than leaving the majority of the protein intact.

Next, we compared the number of potential ORF8 knockouts to potential knockout of other genes in SARS-CoV-2 in our WA dataset. With 13,410 premature stop codons, ORF8 had 14x more stop codons than any other gene (Fig. 2A). The largest genes, ORF1a, ORF1b, and Spike, contained the most large deletions, with >24,000 in each compared to 1517 large deletions in ORF8 (Supplementary Fig. S3A). When normalized by gene length, ORF1a, ORF1b, and spike had a large deletion rate in the range of ORF8 (0.012, 0.023, 0.046 vs. 0.044 per kb per sample respectively) (Supplementary Fig. S3B). Given the necessity of these genes to viral replication, this finding suggests that many potential large deletions could represent missing bases due to poor sequence coverage or amplicon dropout, rather than true deletions. Analyzing the constituent proteins of the ORF1a & ORF1b genes did not show any evidence of a deletion hotspot relative to other SARS-CoV-2 proteins (Supplementary Fig. S3C). When normalized by gene length, ORF7b, M and ORF7a had the highest rate of large deletions (Supplementary Fig. S3B). However, we observe that non-singleton knockout clusters in ORF8 (mean 34.3) are larger on average than non-singleton knockout clusters for all other genes (mean 3.1) ($p = 1.4e{-}05$, Wilcoxon Rank Sum test, one-sided) (Fig. 2B). This result is driven by premature stop clusters (Supplementary Fig. S3C), as we detect no significant difference in large deletion cluster size among genes with the largest deletion cluster sizes: spike (mean 3.3), ORF8 (mean 3.4) and M (mean 4.1) (ANOVA, $p = 0.09$) (Supplementary Fig. S3D). These results are consistent with ORF8 being knocked out more frequently than any other gene in SARS-CoV-2, and the difficulty of identifying large deletions from assembled sequences.

Deleterious mutations are often under purifying selection within a host. If the high rate of ORF8 knockout observed in consensus sequences extended to within-host frequencies, this result would additionally argue against deleterious fitness associated with ORF8 knockout. Therefore, we examined the rate of nonsense mutations in intra-host variants in a subset of 1015 SARS-CoV-2 samples that did not have a consensus-level stop codon, which were sequenced from August-September 2022 in WA (Fig. 3). We defined nonsense intrahost variants as single nucleotide polymorphisms creating a premature stop codon that were present in 1–50% of reads covering that site. Intrahost nonsense variants had to be further supported by at least 10 reads, with a total read of coverage of at least 100 for the site. ORF7b had the highest per codon rate of intrahost nonsense mutations ($8.9 \times 10^{-4}$) followed by ORF8 ($2.4 \times 10^{-4}$). Both genes had elevated intrahost frequencies of nonsense mutations relative to all other genes (ORF7b median = 0.036, ORF8 median = 0.032, other genes median = 0.015) (Fig. 3A). Differences in allele frequency between nonsense mutations in ORF7b/ORF8 and other genes were statistically significant (ORF7b: $p = 5.3 \times 10^{-11}$, ORF8: $p = 4.1 \times 10^{-8}$, Wilcoxon Rank Sum Test) (Fig. 3B). These results are consistent with increased population-level

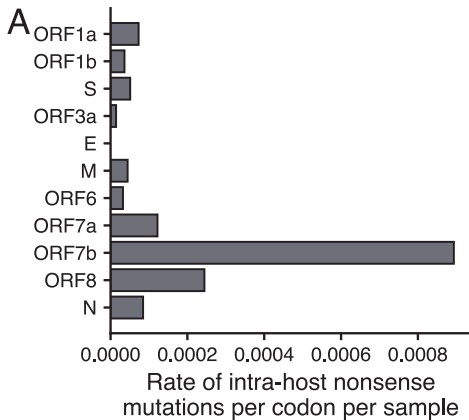

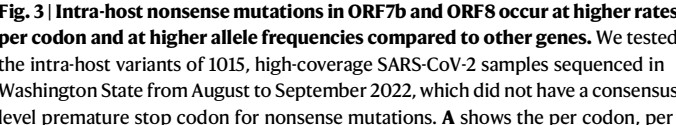

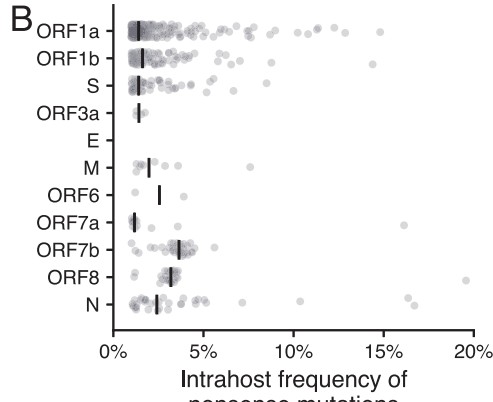

**Fig. 3 | Intra-host nonsense mutations in ORF7b and ORF8 occur at higher rates per codon and at higher allele frequencies compared to other genes.** We tested the intra-host variants of 1015, high-coverage SARS-CoV-2 samples sequenced in Washington State from August to September 2022, which did not have a consensus level premature stop codon for nonsense mutations. **A** shows the per codon, per sample rate of intra-host nonsense mutations in each gene. The frequencies of nonsense mutations observed in intra-host variants are shown in **B** for each gene. Black lines indicate the median frequency. Source data are provided as a Source Data file.

ORF8 knockout and suggest altered within-host selection pressures on both ORF8 and ORF7b.

**Selection pressure acting on ORF8 knockout in a global context**
In WA, we observed that ORF8 is truncated more commonly than any other gene, and clusters containing an ORF8 knockout are larger than clusters with a knockout in any other gene. This result suggests either weakened selection pressure on maintaining ORF8 function relative to other genes or positive selection for ORF8 knockout. The elevated rate and frequency of intrahost nonsense mutations in ORF8 further suggest that either phenomenon extends to within-host evolution. To differentiate between these hypotheses, we applied one of the most widely used measures of selection pressure, $dN/dS$, which compares the ratio of mutation divergence over expectation for both nonsynonymous, or protein modifying, mutations and synonymous mutations, which do not alter the protein's amino acid sequence. Classically, $dN/dS > 1$ is consistent with positive selection as nonsynonymous mutations occur more frequently than synonymous mutations, $dN/dS < 1$ is consistent with negative selection, and $dN/dS \sim 1$ is consistent with neutral evolution. Here, we modified the classic calculation of $dN/dS$ to separately estimate values for missense mutations, which change the amino acid sequence, and nonsense mutations, which introduce a stop codon and result in early truncation of the protein. To mitigate geographic bias, we estimated $dN/dS$ values for each SARS-CoV-2 gene using the publicly available UShER phylogeny, which contains ~7 million SARS-CoV-2 sequences sampled from around the globe (Fig. 4A, Supplementary Fig. S4)[50,51].

In the structural (S, E, M, N) and replicase (ORF1a, ORF1b) genes, we identify strong selection against nonsense mutations, with $dN/dS$ values < 0.01. This result is consistent with these genes being necessary for viral replication. The relative missense estimates for these genes are also largely consistent with expectation. For example, in spike, which has undergone substantial adaptive evolution[14,18], we observe relaxed selection against missense mutation compared to replicase genes ORF1a and ORF1b (with $dN/dS$ values of 0.52, 0.38, and 0.32 respectively). In the accessory genes, which by definition are not necessary for viral replication, $dN/dS$ values for both nonsense and missense mutations are elevated. ORF7a, ORF7b, and ORF8 all have especially high $dN/dS$ values, with missense estimates >1 and nonsense estimates >4.8× that of other genes. Uniquely, ORF8 has values > 1 for both missense and nonsense mutations (1.09 and 1.11 respectively). Classically, $dN/dS$ values of this magnitude are consistent with positive selection; however, caution is warranted when interpreting within-

population $dN/dS$ in terms of selective coefficients[52,53]. Additionally, absolute $dN/dS$ values are sensitive to the substitution matrix used while relative relationships of estimates between genes remain similar regardless of substitution matrix (Supplementary Fig. S5). Comparing across genes, we can conclude that negative selection on mutations in ORF8, ORF7a, and ORF7b are strongly weakened relative to other genes. Our results further suggest positive selection for ORF8 knockout: even with an alternative substitution matrix, $dN/dS$ estimates for ORF8 remained >1 (Supplementary Fig. S5).

To more clearly test for positive selection, we compared success of ORF8 clades with a nonsense mutation to clades with either a synonymous or missense mutation in ORF8 (Supplementary Fig. S6). We found that clades with a nonsense mutation in ORF8 are larger (mean = 77.6, std = 6024.2) and circulate for longer (mean = 11.5 days, std = 35.8) than clades with a synonymous mutation in ORF8 (mean size = 7.0, std = 423.8, mean days = 9.5, std = 28.9). Clades with a missense mutation in ORF8 are also larger on average (mean = 18.5, std = 2482.0) and circulate for longer (mean = 10.3, std = 32.1). For comparison, nonsense mutations in ORF1a and spike are much smaller and circulate for far shorter periods than clades with synonymous mutations.

To statistically quantify these observed differences, we modeled the rate of cluster growth by mutation type as a negative binomial regression of the number of descendants after the mutation was first observed, with an offset for time since observation (Fig. 4B). We found that clusters with nonsense mutations in ORF8 grow 6.3x (95% CI: 5.97–6.52) faster than clusters with synonymous mutations in ORF8. While this approach does not attempt to disentangle the effects of other fitness-impacting mutations which occur downstream of a nonsense mutation, the synonymous cluster growth rate provides a null expectation for comparison. Assuming an absence of epistatic interactions between ORF8 nonsense mutation and other fitness enhancing mutations in SARS-CoV-2, this result suggests that the observed ORF8 knockouts boost viral fitness. This effect is robust to excluding nonsense mutations found in Alpha and XBB descendants, which occurred on highly fit backbones: nonsense mutations still grew 2.5× (95% CI: 2.30–2.76) faster than clusters with synonymous mutations (Supplementary Table S2). Missense mutations in ORF8 grew 1.8× faster (95% CI: 1.77–1.96) than clusters with synonymous mutations in ORF8. This relative fitness benefit could result from either (1) missense mutations disrupting ORF8 function like nonsense mutations, or (2) missense mutations improving fitness by enhancing some aspect of ORF8 function.

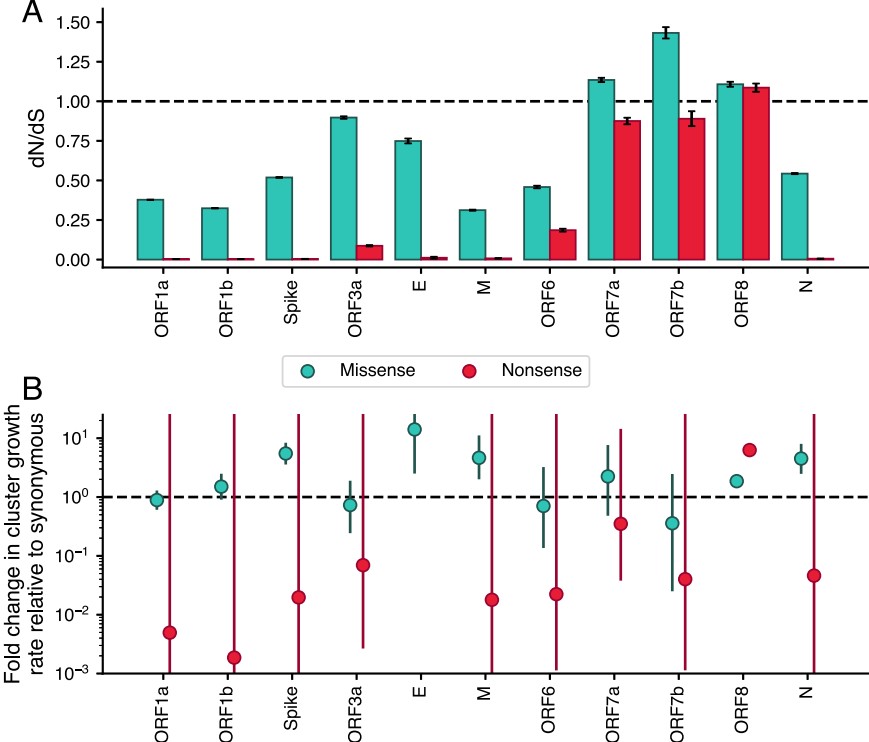

**Fig. 4 | Nonsense mutations in ORF8 result in faster clade growth rates and are more frequently observed than synonymous mutations.** From the global, UShER SARS-CoV-2 phylogeny containing 3,422,473 nodes, we estimated (**A**) *dN/dS* values (shown as bars) and (**B**) the fold change in mutation cluster growth rate relative to synonymous (shown as points) for missense (teal) and nonsense (red) mutations for each gene. Error bars show 95% confidence intervals for each calculation. For *dN/dS*, confidence intervals were calculated by 10,000 bootstrap iterations across all nodes in the tree. E did not have enough nonsense mutations to calculate a cluster growth rate. Source data are provided as a Source Data file.

For all other genes, clusters with nonsense mutations grew at a reduced rate (0.07× on average) relative to clusters with synonymous mutations. These differences were not statistically significant, likely due to the very few number of nonsense mutation clusters observed in most genes resulting in wide CIs (Fig. 4B). For example, in spike and ORF1a, 0.035% and 0.017% of mutations were nonsense respectively. Comparing cluster growth rates assumes mutations occurring and being sampled, so rates will be less sensitive for detecting clusters with mutations under strong negative selection. For ORF7a and ORF7b, which had elevated counts of nonsense mutations, the absence of a difference in cluster growth rate between nonsense and synonymous mutations could be consistent with neutral evolution in these genes.

The difference in growth rate with a nonsense mutation in ORF8 is similar to that of increase in growth rate for a missense mutation in spike: 5.5× (95% CI: 3.57–8.38). Since many missense mutations in spike are associated with fitness gains, this finding also suggests positive selection for ORF8 knockout. We did not observe a significant difference in growth rate for missense mutations in ORF1a (0.88, 95% CI: 0.607–1.29). However, if mutations are split out into mutations with positive fitness previously inferred by a hierarchical logistic regression model[19] versus other missense mutations, clusters with fitness-associated missense mutations grew 4.3× faster (95% CI: 2.08–10.11) than synonymous mutation clades while cluster with other missense mutations grew 0.43× slower (95% CI: 0.29–0.64×) relative to synonymous (Supplementary Fig. S7). This is consistent with predominantly negative selection on ORF1a, but occasional mutations under positive selection. We observed a similar split of cluster growth rates for ORF8 and spike when classifying mutations by type and previously inferred positive fitness. These results suggest that our cluster growth analysis is in agreement with previous work (Supplementary Fig. S7)[19].

To further test if increased fitness for ORF8 knockout was driven by specific clades, we estimated ORF8 *dN/dS* rates split out by each Nextstrain clade, for each clade larger than 500 clusters (Supplementary Fig. S8A). Most clades had wide confidence intervals, with *dN/dS* rates for nonsense mutations indistinguishable from 1 for 17 clades. An additional eight clades – 19A, 20A, 20B, Alpha, Gamma, 21C, 21F, and 21H – had rates significantly greater than 1. Only four clades – 21K, 20G, 22D, 20F – had rates significantly less than one, which suggests that mutations are well-tolerated across the entire tree. A one-sided Wilcoxon Signed Rank test comparing if per-clade point estimates for *dN* were larger than *dS* was significant for missense mutations but not nonsense mutations ($p = 0.0057$ & $p = 0.24$ respectively).

Due to the massive differences (over 1000×) in the ratio of largest nonsynonymous cluster over largest synonymous cluster associated with high dispersion of cluster size and the relatively few mutations per clade, we were not able to reliably estimate growth rate advantages per clade. Instead, we compared the ratio of the geometric mean of nonsynonymous cluster size and geometric mean of synonymous cluster size for each clade (Supplementary Fig. S8B). Since each clade covers a smaller time window than the entire tree, the size bias from different cluster start times is minimized. Confidence intervals generated by bootstrapping across clusters were wide, suggesting we had limited power to determine if nonsynonymous clusters were larger than synonymous clusters. In 21J, 20A, Alpha, and 22C nonsense clusters were significantly larger on average than synonymous clusters; however, no clades had nonsense clusters significantly smaller on average than synonymous clusters. A one-sided Wilcoxon Signed Rank test comparing if per-clade point estimates for nonsynonymous cluster growth ratios were larger than synonymous cluster growth ratios was not significant for either missense mutations or nonsense mutations ($p = 0.88$ & $p = 0.39$ respectively).

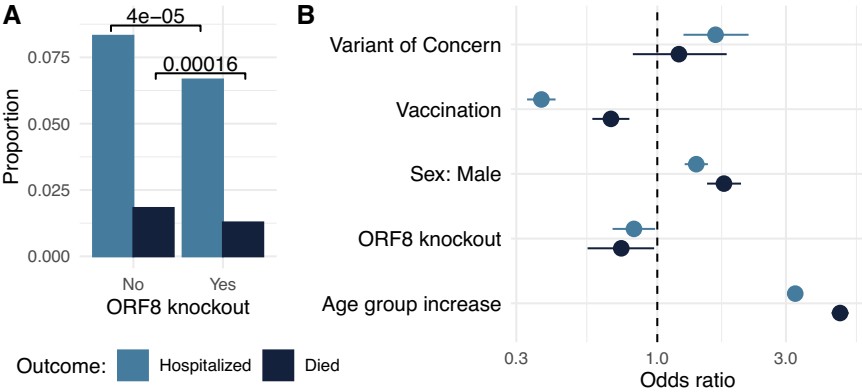

**Fig. 5 | ORF8 knockout is associated with reduced clinical severity of COVID-19.** **A** Proportion of individuals with severe COVID-19 outcomes stratified by virus infection with and without ORF8 knockout. *P*-values for α = 0.05 from a two-sided, Fisher's exact test are shown. **B** Odds ratios from a generalized linear model of clinical outcomes for variants of concern (Alpha, Beta, Gamma or Delta), vaccination status, assigned male sex at birth, ORF8 knockout, and increasing age. Points show the estimated odds ratio, and error bars show 95% confidence intervals. Severe COVID-19 outcomes are hospitalization (light blue) and death (dark blue). Analysis was limited to pre-Omicron lineages due to reduced clinical severity and loss of vaccine efficacy associated with Omicron variants (hospitalization regression: *n* = 25,531, death regression: *n* = 49,468). Source data are provided as a Source Data file.

Combining both *dN/dS* estimates and cluster size ratios for each clade, we see that 6 clades are differentiated from other clades by their elevated measures for nonsense mutations in both metrics, even if they failed to achieve significance levels (Supplementary Fig. S8C). These clades span a variety of time windows, which suggests that a time-correlated trend, such as development of high-levels of population immunity, does not drive the increased counts and success of ORF8 knockouts. In contrast, we identify only three clades with reduced *dN/dS* rates and smaller cluster sizes for nonsense mutations on average. Overall, these results suggest that the transmission advantage of ORF8 knockout may vary somewhat across clades, but Alpha and XBB alone are not the only clades with increased growth advantages.

## Clinical impact of ORF8 knockout

Previous analysis found a large deletion that knocked out ORF8 and truncated ORF7b to be associated with reduced clinical severity[22]. Here, we extend this analysis to the clinical impact of any ORF8 knockout, due to large deletion or premature stop codon, by linking SARS-CoV-2 sequences with clinical outcomes recorded in the Washington Disease Reporting System. Given the reduced clinical severity and loss of vaccine efficacy associated with Omicron variants, we restricted our analysis to pre-Omicron lineages. Supplementary Table S3 outlines the general characteristics of our study population stratified by infections with and without an ORF8 knockout. While 8.3% (*n* = 1906/22,928) of individuals infected by SARS-CoV-2 with intact ORF8 were hospitalized, only 6.7% (*n* = 383/5746) of individuals infected by virus with ORF8 knocked out were hospitalized (*p* = 3.1 × 10⁻⁵, Fisher's exact test) (Fig. 5A). Similarly, 1.8% (*n* = 910/49,912) of individuals infected by virus with intact ORF8 died due to SARS-CoV-2 compared to 1.3% (*n* = 129/10,089) of individuals infected by virus with an ORF8 knockout (*p* = 8.2 × 10⁻⁵, Fisher's exact test) (Fig. 5A). However, ORF8 knockouts have not been distributed evenly across time in Washington State (Fig. 1A), and clinical severity of SARS-CoV-2 has varied temporally with changing age circulation patterns, the rollout of vaccines, accumulation of natural immunity in the population, medications, and viral evolution.

In a general linear model adjusting for week of collection, variant of concern, vaccination status, sex at birth, and age group, we found a 0.82 (95% CI: 0.68–0.98) odds ratio of hospitalization in infections containing an ORF8 knockout compared to infections without the knockout (Fig. 5B). The odds of death when infected by virus with an ORF8 knockout was also reduced (Odds ratio: 0.73, 95% CI: 0.55–0.97).

In both regressions, vaccination was associated with reduced clinical severity while male sex and increase in age group were associated with worse clinical outcomes. When compared to other SARS-CoV-2 lineages, variants of concern – Alpha, Gamma, Delta, or Beta lineages – were independently associated with increased odds of hospitalization but not with odds of death. While the effect sizes estimated are barely significant, power analysis identifies only 22% power to identify a significant effect for hospitalization and 75% power to find an effect for death.

Given the difficulty of accurately calling large deletions in ORF8, we tested the robustness of our effects by the size of cluster required to define a knockout. To calculate cluster size, we built three additional maximum likelihood phylogenies enriched for ORF8 knockouts in WA one for Delta, one for Alpha, and one for other non-Omicron lineages. These breakdowns were chosen such that all ORF8 knockouts sequenced in WA could be placed in an appropriate phylogenetic context of at least 75% background sequences. We then reconstructed parsimony clusters for ORF8 knockout (see above and Methods). We found that both effect size for ORF8 knockout and model Akaike information criterion (AIC) minimally changed with various cluster sizes required to define a knockout (Supplementary Fig. S9). This demonstrates that the clinical effect is robust to inaccurate identification of ORF8 knockout due to large deletion.

## Discussion

The SARS-CoV-2 pandemic has been characterized by a high rate of evolution as fitness enhancing mutations, primarily in spike, have repeatedly swept globally. Here, we explored the selection pressures underlying a surprising and repeated sweeping mutation pattern: ORF8 knockout.

Examining ORF8 knockout across time in a Washington State, we found that while knockout spread widely in Alpha and XBB.1 descendant lineages, it also occurred at a low frequency on many other viral backbones due to both large deletions and premature stop codons (Fig. 1). This finding is consistent with other reports of large deletions encompassing ORF8 circulating in other parts of the globe[21,23–27]. While knockout is observed in other genes in Washington State[54], we find that ORF8 has more premature stops than any other gene; knockout clusters with ORF8 grow larger than knockout clusters for any other gene, and the rate and frequency of intra-host nonsense mutations in ORF8 are elevated (Figs. 2 and 3). At a global level, we estimate a higher than expected number of nonsense and missense mutations in ORF8 (Fig. 4). Nonsense mutations in

ORF8 show the highest nonsynonymous over synonymous divergence for any gene in SARS-CoV-2.

Together these results recommend rejecting our first hypothesis: that ORF8 knockout is deleterious to SARS-CoV-2 fitness and fixation was driven by hitchhiking on the back of other fitness enhancing mutations. The $dN/dS = 1.1$ estimates suggest that ORF8 knockout is due to positive selection, though estimates in a single evolving population should be interpreted with caution[52,53]. We next modeled the rate of mutation cluster growth rates and found that clusters with nonsense mutations grow roughly 6x faster than synonymous mutations in ORF8. Excluding stop mutation clusters present in XBB and Alpha, we still find that nonsense mutations in ORF8 grow 2.5× faster than synonymous. These values are comparable to the improvement in cluster growth rates by missense mutations in spike over synonymous and further suggest that ORF8 knockout boosts viral fitness.

This conclusion is broadly consistent with other estimates of the fitness effects of SARS-CoV-2 mutations. In an updated run of the fitness model presented in Obermeyer et al., numerous stop mutations in ORF8 are estimated to boost fitness[19]. Bloom and Neher only find evidence of relaxed purifying selection on ORF8 and other SARS-CoV-2 accessory proteins. However, the difference between their fitness effects and our $dN/dS$ estimates for ORF8 are consistent with the limited correlation between fitness effects and $dN/dS$ estimates they observed. As count-based methods, both approaches are underpowered to identify positive selection since they only explore how often mutation occurs, not how large clades get once mutation occurs[20]. We also found evidence of relaxed purifying selection in SARS-CoV-2 accessory proteins as we estimated high nonsense and missense $dN/dS$ values for ORF7a and ORF7b, in addition to ORF8 (Fig. 4). However, unlike ORF8, we did not find evidence for a growth rate advantage for ORF7a or ORF7b knockouts (Fig. 4). This clarifies previous reports of ORF7a and ORF7b deletions[21,24,25,54], and our observation that ORF8 knockout clusters grew larger than for any gene. It also suggests that ORF7a and ORF7b could be deleted in future SARS-CoV-2 evolution. However, ORF8 may be deleted more quickly, due to the fitness benefit associated with ORF8 knockout.

Consistent with previous analysis[22], we found evidence of reduced clinical severity with ORF8 knockout (Fig. 5). This observation might help explain why Alpha had reduced clinical severity compared to the Beta, Gamma, and Delta variants of concern[47]. It also highlights the importance of studying the clinical impact of SARS-CoV-2 evolution; genetic changes in the virus can have different effects on clinical severity.

Our results imply that SARS-CoV-2 genomic surveillance should include detection of ORF8 knockout going forward. Currently, much of circulating SARS-CoV-2 has ORF8 knocked out; however, when ORF8 knockout arises on a viral backbones with intact ORF8 expression this suggests a transmission advantage. Conversely, rescue of ORF8 protein expression could increase the clinical severity of COVID-19 infections, though the effect may be small. Knockout due to point mutation or frameshifts can be readily detected from assembled viral genomes. To detect large deletions, assembled genomes can be screened for long stretches of N's, which will result in numerous false positives, or raw reads and sequence alignment maps can be screened for ORF8 deletions earlier in genome assembly pipelines.

A dispensable ORF8 and increased viral replication speed due to a shortened genome cannot explain positive fitness effects associated with premature stop codons. Host restriction factors, which have well established impacts on the evolution of other viruses could play a role[55]. The only other coronavirus with an ORF8, SARS-CoV, also had ORF8 knocked out, suggesting a repeated evolutionary pattern[30]. Alternatively, recent work by Kim et al identifies an intriguing biological mechanism underlying positive selection for ORF8 knockout and the timeline for knockout to sweep[56]. Their study finds that ORF8 covalently interacts with spike at the endoplasmic reticulum, reducing onward transport of spike to the cell membrane and incorporation into virus particles. Presence of ORF8 is associated with less spike in pseudovirions. Less spike in virions and on the cell surface might improve viral fitness within the individual by providing another mechanism for SARS-CoV-2 to avoid the host immune response. However, when ORF8 is knocked out, more spike in virions might improve viral fitness at a transmission level by making it easier to establish infection. While we observed an elevated within-host rate and frequency of ORF8 nonsense mutations, our sequenced samples were likely from acute infections where the relative effect of immune pressure may matter less than in chronic infections which are a hypothesized source of variants of concern.

Given the magnitude of the transmission advantage estimated for ORF8 knockout, it is puzzling that ORF8 knockouts have not been fixed and have only spread globally in Alpha and XBB subclades. While we identified some heterogeneity across clades in ORF8 $dN/dS$ rates and nonsense-synonymous cluster size ratio, increased mutation counts and growth advantage were not limited to Alpha and XBB. Thus, although viral backbone could modulate fitness effects to some degree, it does not explain the few occurrences of ORF8 knockout reaching appreciable global frequencies. ORF8 knockout may be a classic example of clonal interference, especially considering the low probability of introducing a gene knockout and the high fitness boosts of mutations in other parts of the genome. For example, just as it appeared that ORF8 knockout from XBB descendants might globally fix, BA.2.86 viruses outcompeted XBB, dropping ORF8 knockout frequencies[57]. The tug of war that Kim et al propose between within-host fitness and between-host fitness could also contribute to the lack of ORF8 knockout fixation. ORF8 nonsense mutations are absent from chronic infection-associated mutations[58]. Within chronic infections, intact ORF8 could provide a necessary edge to evade the host immune system. The hypothesized disproportionate contribution of chronic infections to global SARS-CoV-2 evolution could prevent ORF8 knockout fixation[59–63].

## Methods

### Calling gene knockouts

On April 24, 2023, we downloaded all 149,547 SARS-CoV-2 sequences from GISAID collected in WA through March 31, 2023[64]. Sequences were called as having a potential knockout in a gene if either 30 consecutive nucleotide bases in the coding sequence for that gene were gap characters or N's, or if the predicted protein coding sequence was more than 10 codons shorter than the reference protein, due to a premature stop codon from a nonsense or frameshift mutation. With short-read sequencing and reference-based genome assembly as are commonly used in SARS-CoV-2 sequencing pipelines[65], large deletions will show up as long stretches of N's; however, long stretches of N's could also represent poor sequence quality or amplicon dropout. To limit bias from poor sequencing quality, samples had to have a genome coverage of at least 95%, or no more than 1495 missing bases. In ORF8, we excluded calling large deletions between bp 27809-27854 in samples with a C27807T mutation as this mutation was associated with amplicon dropout in the ARTIC v4 sequencing primers[66]. We considered alternative cutoff lengths for knockouts, balancing between wrongly calling short, likely functional deletions as gene knockouts and preferentially maximizing or minimizing the number of knockouts in any one gene (Supplementary Fig. S1).

### Sanger sequencing & PCR validation of large deletions

Sequencing of remnant clinical specimens at UW Virology Lab was approved by the University of Washington Institutional Review Board with a waiver of informed consent (protocol STUDY00000408). We performed screening by PCR and Sanger sequencing on a subset of samples to determine whether long strings of ambiguous bases (Ns) in ORF7 and ORF8 were the result of deletions rather than amplicon

dropout. All 9998 samples sequenced by the University of Washington as of May 2021 were screened, and 120 were found to have sequences with contiguous strings of Ns (>266 bp) from ORF7a through ORF8. Of these 120, 89 were determined to have sufficient volume and sample quality for PCR and Sanger Sequencing. Total nucleic acid extraction was done on the MagNA Pure 96 instrument (Roche) with 200 μL sample input and 50 μL elution volume. Amplification was performed with SuperScript-III One-Step RT-PCR kit (Invitrogen, Waltham, MA, USA) and primers designed flanking the deletion region, beginning in ORF7a (forward: GGCACTGATAACACTCGCTAC) through the beginning of the N-gene (reverse: GAGGGTCCACCAAACGTAATG). Thermocycling conditions were as follows: 55 °C for 30 min, 94 °C for 2 min, and 35 cycles of 94 °C for 30 s, 58 °C for 30 s, and 68 °C for 1 min. A final extension step was included at 72 °C for 5 min. Reactions were cleaned with SPRI Ampure beads (Beckman Coulter, Brea, CA, USA) at a 0:0.65 volumetric ratio and eluted to 40 μL. Flash gel electrophoresis (Lonza, Basel, Switzerland) was performed to confirm successful PCR and for preliminary deletion calling. Samples were diluted and sent for Sanger sequencing on ABI's Prism 3730xl DNA analyzer (Genewiz, Seattle, WA, USA) with the designed primers. Consensus sequences were aligned against NC_045512.2 to confirm the presence of the deletions.

### Phylogenetic reconstructions

To determine if potential gene knockouts might be part of the same transmission cluster, we built a maximum likelihood phylogeny of SARS-CoV-2 enriched for WA sequences. We used the Nextstrain pipeline[67] to align sequences to Wuhan-Hu-1/2019 (genbank accession MN908947.3) using nextalign[68], to reconstruct a maximum-likelihood phylogeny using IQ-TREE[69], to estimate molecular clock branch lengths using TreeTime[70], and to infer nucleotide and amino acid substitutions across the phylogeny. For IQ-TREE, we specified a GTR substitution model, 10 initial parsimony trees, and four unsuccessful iterations to stop; for TreeTime, we used a substitution rate of 0.008 with a standard deviation of 0.004. The input to the pipeline was a focal ~10,000 sequences collected in WA and an additional ~10,000 contextual sequences, 5000 sequences from the rest of the United States and 5000 sequences from other countries. For each geographic region, all sequences were sampled evenly across time from the beginning of the SARS-CoV-2 pandemic through March 2023. WA sequences were downloaded from GISAID (described above), and contextual SARS-CoV-2 sequences sampled elsewhere in the United States and around the globe prior to March 21, 2023 were downloaded from GISAID on July 27, 2023. We used the default settings for the Nextstrain SARS-CoV-2 pipeline (https://github.com/nextstrain/ncov/tree/master) to filter this dataset, except we increased genome coverage ≥95% to minimize large deletions that represent poor sequence coverage. The pipeline additionally excludes samples with incomplete dates, samples with >20 deviation from the molecular clock rate, samples with >5 private reversions, and samples with more than 6 private mutations in a 100-nucleotide window. The final phylogeny contained 16,268 sequences. This phylogeny is available to view at: https://nextstrain.org/groups/blab/ncov-orf8ko/WA/20k.

Also using the Nextstrain pipeline, we built three additional clade-specific phylogenies enriched for sequences with potential large deletions in ORF8 sampled in WA. We built one for Alpha, one for Delta, and one with all other, pre-Omicron SARS-CoV-2 lineages. These numbers were chosen such that every sequence collected in WA with a potential knockout of ORF8 was contained in a phylogeny. The input to the pipeline was all potential ORF8 KO's in that SARS-CoV-2 clade, no more than 5000 sequences. For context, we included 5000 additional WA sequences without ORF8 KO's, 5000 sequences from the rest of the United States, and 5000 sequences from around the globe. Contextual samples were evenly temporally sampled from each geographic region. The pipeline filtered samples as above, and the final trees respectively included 18,350, 14,908, 12,050 sequences.

These phylogenies are available to view at: https://nextstrain.org/groups/blab/ncov-orf8ko/WA/Alpha, https://nextstrain.org/groups/blab/ncov-orf8ko/WA/Delta, and https://nextstrain.org/groups/blab/ncov-orf8ko/WA/other.

### Knockout clustering

Knockout clusters were called using clustering methods appropriate for each knockout type. Large deleted segments can only be recovered via recombination, and for the purpose of this analysis we considered this an unlikely event. Therefore, clusters of large deletions were reconstructed using the Camin-Sokal parsimony algorithm[71], which is a unidirectional parsimony clustering algorithm. We considered sequences to be part of the same deletion cluster if all their common ancestor nodes and all descendant nodes shared a deleted region of at least 30 nucleotides. Premature stop codons introduced by nonsense or frameshift mutations can be removed by back mutation. Thus, knockout clusters due to premature stops were called using the Fitch parsimony algorithm, which allows for back mutation[65,72]. Samples were considered as part of the same knockout cluster if all their common ancestor nodes contained the same premature stop codon.

### Intrahost analysis

We examined the rate and frequency of nonsense mutations in intrahost single nucleotide variants in 1300 SARS-CoV-2 samples sequenced by the University of Washington from August to September 2022 as part of a genomic surveillance program. Sequencing of remnant clinical specimens at UW Virology Lab was approved by the University of Washington Institutional Review Board with a waiver of informed consent (protocol STUDY00000408). Nasopharyngeal, nasal, or oropharyngeal swabs with PCR cycle threshold <31 were randomly selected and sequenced as described previously[73]. Briefly, after RNA extraction, library preparation was performed using the Illumina COVIDseq protocol with ARTIC v4.1 primers (Integrated DNA Technologies). Prepared libraries were pooled and sequenced on an Illumina Novaseq6000 instrument using a 2 × 150 read format targeting at least 1 million reads per sample. Genome assembly was performed using a custom pipeline (https://github.com/greninger-lab/covid_swift_pipeline) which performs trimming to remove adapters and low quality regions, primer clipping, variant calling, and consensus genome generation. We excluded 55 samples due to inadequate coverage (>10% N's or <7419 reads, which was two standard deviations under the mean coverage) or poor amplification (>25% of reads trimmed). We excluded an additional 230 samples with a consensus-level premature stop in any gene to avoid biasing rates of intrahost nonsense mutations. In the remaining 1015 samples, we required all intrahost nonsense variants to have ≥1% frequency with a variant coverage of 10× and a total position coverage of 100×.

### Calculating dN/dS

For selection analyses, to mitigate geographic bias, we downloaded the publicly available, mutation-annotated, SARS-CoV-2 UShER tree[50,51] on May 1, 2023 from: http://hgdownload.soe.ucsc.edu/goldenPath/wuhCor1/UShER_SARS-CoV-2/2023/05/01/. For our analyses, we trimmed this tree to remove sequences without associated collection dates using matUtils 0.6.2 (https://usher-wiki.readthedocs.io/en/latest/matUtils.html#).

We calculated the expected number of synonymous, missense, and nonsense sites for each gene by multiplying each base in the coding sequence by the substitution rates for that base previously inferred for SARS-CoV-2. We then summed together expected mutations by mutation type for each gene. We considered two sources for inferred substitution rates: (1) substitution rates calculated from the 4-fold degenerate sites within SARS-CoV-2 using the global UShER phylogeny[20], and (2) a maximum likelihood substitution matrix inferred early in the pandemic from 36 SARS-CoV-2 genomes[74]. In the main

text, we present results from the first source (Fig. 3A), but include results for all genes for both substitution matrices in the Supplement (Supplementary Fig. S4).

We calculated the observed number of synonymous, missense and nonsense mutations in the UShER phylogeny by classifying the reconstructed mutations at each node by their gene and mutation type. We generated the divergence for each mutation type for each gene by dividing the number of observed mutations by the expected number of sites. For missense and nonsense mutations, dN/dS were calculated by dividing the divergence for those respective mutation types by the synonymous divergence. While the signal strength observed with an UShER tree as opposed to a smaller global phylogeny is weakened since the number of segregating polymorphisms in the population is greatly increased[52], our analysis focused on comparing the relative differences in dN/dS by mutation type across genes rather than the absolute magnitude of dN/dS values.

## Identifying mutation clusters

To compare cluster size and circulation time by mutation type for SARS-CoV-2 genes, we classified point mutations on each node in the SARS-CoV-2 UShER tree as synonymous, missense, or nonsense. Nodes containing multiple mutations in the same gene were excluded from the analysis. Cluster size represented the total number of tips descended from that node and days of circulation was calculated from the latest to the earliest date at which a descendant tip was sampled. By this definition, clusters may contain nested clusters with mutations in that gene that could contribute to their success. However, we chose this definition as using non-nested clusters with a single mutation type per gene truncates signals of positive selection because cluster size is maxed out as a function of the molecular clock rate and length of the gene.

## Modeling mutation cluster growth rate

Using negative binomial regression, we can model number of additional descendants, given we observed a mutation as:

$$cluster\ size \sim NegBinom(\mu_{mutationtype}, \theta) \quad (1)$$

Where $\mu_{mutation\ type}$ is the expected number of descendants of a given mutation and $\theta$ is the over-dispersion relative to Poisson distribution. Since each cluster can grow for a different period of time, we can model number of additional descendants per unit time since the mutation was first observed by:

$$\log\left(\frac{\mu_{mutationtype}|mutation}{time\ since\ mutation}\right) = \beta'_0 + \beta'_1 \times mutation\ type \quad (2)$$

$$\log(\mu_{mutationtype}|mutation) = \beta'_0 + \beta'_1 \times mutation\ type + \log(time\ since\ mut) \quad (3)$$

The estimated parameters $\beta'_1$ then correspond to the log fold increase in the cluster size growth rate for a given mutation.

Likelihood ratio test for negative binomial regression compared to Poisson regression indicated that the negative binomial model was more likely due to overdispersion of cluster growth rate ($p = 0$). The negative binomial model was fit in R using the MASS package (https://cran.r-project.org/web/packages/MASS/index.html), and the Poisson model was fit in R using the GLM package (https://cran.r-project.org/web/packages/glm2/glm2.pdf), specifying family = Poisson. We fit negative binomial models separately for each gene with mutation types split out by synonymous, missense, and nonsense. For spike, ORF1a, and ORF8, we fit additional models with mutations split out by type and fitness advantage previously inferred by a hierarchical logistic regression model[19].

## Clade-level analysis

We labeled nodes in the UShER phylogeny with Nextstrain clades by passing clade labels from tip to parent nodes using a backwards traversal algorithm. If a node had multiple potential clade labels, the clade first identified was passed up to the parent node, i.e. 19 A preceded 19B. We calculated ORF8 dN/dS for each clade with more than 500 samples using the same method applied to the entire phylogeny, except subsetting the tree to only nodes in that clade.

To test the growth advantage of nonsynonymous mutations over synonymous mutations for individual clades, we calculated the ratio of the geometric mean size of nonsynonymous clusters divided by the geometric mean size of synonymous clusters. We calculated the ratio split out by missense and nonsense mutations for each clade with more than 500 samples. Confidence intervals were generated by bootstrapping 10,000 times across nodes. We chose this approach, rather than modeling cluster growth rate, because it was more robust to biases from a single cluster, given the much smaller number of clusters in each clade relative to the entire tree. Since clades encompass smaller time windows than the entire phylogeny, average cluster size is less biased by different cluster starting times.

## Clinical analysis

Under Washington State IRB Exempt Determination 2020-102, age, sex, hospitalization, death, and vaccination history was provided by the Washington Department of Health from the Washington Disease Reporting System for individuals with linked sequenced SARS-CoV-2 samples from June 1, 2020 through July 31, 2022. We limited the clinical analysis to pre-Omicron lineages since Omicron was associated with reduced clinical severity and loss of vaccine efficacy.

We used a Fisher's exact test to compare the number of individuals who were hospitalized or died due to SARS-CoV-2 infection by presence of an ORF8 knockout in their sequenced sample.

To estimate the impact of ORF8 knockout on clinical outcomes of hospitalization and death, we used a multivariate logistic regression:

$$logit(P_i) = \beta_0 + \Sigma\beta_j x_{i,j} + \epsilon_i \quad (4)$$

Where $P$ is the probability of hospitalization or death, $\beta$ is the coefficient of the predictor variable, $x$ is the predictor variable, and $\epsilon$ is the residual error. Predictor variables were: ORF8 knockout (binary variable), sex assigned at birth (binary variable), age group (discrete variable), vaccinated (binary variable, variant of concern (binary variable), and week of collection (categorical variable). Only sex assigned at birth: Male or Female were included in the model as there were too few Other samples to estimate a coefficient. Age groups were 0–4yo, 5–17yo, 18–44yo, 45–65yo, 65–79yo, and 80+yo. Variants of concern were Alpha, Beta, Delta, and Gamma lineages as designated by the World Health Organization[75]. Individuals were considered to be vaccinated if two weeks passed since any COVID-19 vaccination. The model was fit in R using the GLM package (https://cran.r-project.org/web/packages/glm2/glm2.pdf).The package Glm was used to conduct the logistic regression. To identify the power to determine a significant effect of ORF8 knockout on death, we used the pwr package in R. Specifically, we used the power test for the general linear model "f2.test" to estimate our power to identify the effect estimated by the general linear model. We calculated Cohen's $f^2$ for ORF8 knockout as previously described[76] by the below equation:

$$f^2 = \frac{R^2_{AB} - R^2_A}{1 - R^2_{AB}} \quad (5)$$

where $R^2_{AB}$ is the McFadden's R-Squared value for the model with all coefficients, including ORF8 knockout, and $R^2_A$ is the McFaden's R-Squared value for the model with all coefficients, except ORF8 knockout.

Given the challenges of classifying ORF8 knockouts due to large deletions, we explored robustness of our model fit and effect size by the criteria required to classify an ORF8 knockout. We introduced the additional criteria that an ORF8 knockout had to cluster with some threshold number of other ORF8 knockout samples in order to be considered a true knockout. We then computed Akaike Information Criterion and the odds ratio of ORF8 knockout for outcomes of hospitalization and death using thresholds from 0 to 50.

## Reporting summary

Further information on research design is available in the Nature Portfolio Reporting Summary linked to this article.

## Data availability

All GISAID metadata and sequences used to identify knockouts in ORF8 in WA and to build WA focused phylogenies in analyses are available at: gisaid.org/EPI_SET_230921by. Sequencing reads from the intrahost analysis have been deposited on the SRA under bioproject number PRJNA738869: https://ncbi.nlm.nih.gov/bioproject/PRJNA738869 and PRJNA610428: https://ncbi.nlm.nih.gov/bioproject/PRJNA610428. The list of samples whose intrahost variants were analyzed and the identified variants after quality control are available at: https://github.com/blab/ncov-orf8/blob/main/intrahost/. The UShER phylogeny used in selection analyses is available at: http://hgdownload.soe.ucsc.edu/goldenPath/wuhCor1/UShER_SARS-CoV-2/2023/05/01/.

Clinical data was provided by the Washington Department of Health. To protect patient privacy, the full dataset is not publicly available per the terms of the data use agreement. However, a subset of the data variables (vaccination status, sex assigned at birth, ORF8 knockout, variant of concern, hospitalization, death) are available at: https://github.com/blab/ncov-orf8/blob/main/data/clinical_subset.tsv. The full data is available from the authors upon reasonable request and permission of the Washington State Department of Health. Source data for all figures are provided with this paper. Source data are provided with this paper.

## Code availability

All code used in this analysis is publicly available at: https://github.com/blab/ncov-orf8[77]. Code was written in both Python 3.10.9 and R 4.1.2.

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

## Acknowledgements

We would like to thank Allison Thibodeau, Topias Lemetyinen, Allison Warren, Cameron Ashton, Emily Nebergall, Peter Gibson, and Sarah Menz for their work throughout the pandemic linking SARS-CoV-2 sequences in Washington State to the Washington Disease Reporting System. We also gratefully acknowledge all data contributors, i.e., the Authors and their Originating laboratories responsible for obtaining the specimens, and their Submitting laboratories for generating the genetic sequence and metadata and sharing via the GISAID Initiative, on which this research is based. C.W. is funded by Achievement Rewards for College Scientists. T.B. is an Investigator of the Howard Hughes Medical Institute. K.K. is also supported by Howard Hughes. M.F. is supported by the NSF Graduate Research Fellowship Program under Grant No. DGE-1762114. The Scientific Computing Infrastructure at Fred Hutch is supported by NIH grants S10-OD-020069 and S10-OD-028685. Funding for Washington Department of Health data collection was provided by Centers for Disease Control and Prevention (CDC) ELC EDE.

## Author contributions

C.W. designed and performed all analyses. K.K. provided key support for selection analyses. G.A.P., N.B. identified and confirmed initial ORF8 deletions. L.A.F. and L.M.T. were critical to clinical data collection and curation. L.A.F. and H.N.O. provided key support for clinical analyses. F.A. and C.Y. linked clinical metadata to sequences. M.F. provided key support for cluster growth rate analysis. A.C., H.N.O. oversaw clinical data collection and curation. A.L.G., H.N.O., P.R., and T.B. oversaw the study. C.W. and T.B. wrote the manuscript. All other authors edited the manuscript.

## Competing interests

A.L.G. reports contract testing from Abbott, Cepheid, Novavax, Pfizer, Janssen and Hologic, research support from Gilead, salary and stock grants for an immediate family member, outside of the described work. All other authors declare no competing interests.
