## [Peer Review File · Nature Communications]

Positive selection underlies repeated knockout of ORF8 in SARS-CoV-2 evolutionReviewer #1 (Remarks to the Author):

The authors investigate the repeated emergence of ORF8 knockout during SARS-CoV-2 evolution, which is primarily associated with Alpha and XBB variants (though across the other lineages/variants prevalence is ~3%). Given the frequency of ORF 8 knockouts, which now dominates current strains, the authors set out to test whether ORF 8 knockout is mildly deleterious or neutral and has hitchhiked with other adaptive mutations in the virus genome. They use two datasets, the global dataset collated to reconstruct the Usher phylogeny, and a local dataset from Washington state, where sequences with ORF 8 knockouts are sampled more evenly through time. To understand the role of selection in emergence and fixation of ORF8 knockouts, they undertook dN/dS analysis and examined the success of clades with ORF knockouts by calculating the cluster size. These findings were compared to key genes, which are necessary for viral replication, some of which have undergone adaptive evolution (e.g. spike).

Overall, I find the main conclusion that ORF8 knockouts are under positive selection and provide a fitness advantage to the virus compelling, but I wonder how robust these results are when the dataset is split by variants? Does the growth advantage and elevated nonsynonymous rates of ORF 8 knockouts hold across all variants, or just Alpha and XBB variants, where this mutation dominates? Are there differences in persistence times of the ORF 8 knockout clades between the two variants? Given that the virus would be associated with different population sizes (i.e. infections) and different levels of population immunity when Alpha and XBB variants emerged, I expect there would be some differences, which could provide further insight into the evolutionary impact of ORF 8 knockouts.

I am less certain about their results indicating ORF 8 knockouts are associated with reduced severity since the odd ratios though < 1 are very close to 1. To me, this does not appear to be strong evidence for reduced severity, particularly to warrant screening for intact ORF8.

Furthermore, the discussion is quite limited, which mainly focuses on reiterating the key results. For example, I would like to see some discussion of why ORF 8 knockouts appear to dominate during Alpha and XBB variants, but not other lineages/variants? Why might ORF 8 knockout provide a transmission advantage in these two variants?

Reviewer #2 (Remarks to the Author):

The study by Wagner and colleagues presents a succinct, well-described and compelling argument that global evolution of SARS-CoV-2 has included repeated loss of ORF8 function, and that ORF8 loss consistently confers evolutionary advantage. Although the frequent loss of ORF8 has been remarked upon by others, to my knowledge this is the first comprehensive synthesis of the available evidence. This finding is supported by a rigorous analysis of relative growth rates of viral clades that have lost ORF8 expression either by nonsense mutation or via large deletions. The authors demonstrate that ORF8-knockout clades have a transmission advantage over other circulating viral clades, even when clades with the highest transmission advantage (such as Alpha) are excluded. A strength of the study is the inclusion of datasets at both the regional level (Washington State surveillance Feb 2020 - March 2023) and the global (Usher phylogeny). Also commendable are sensitivity analyses and validation work, including additional Sanger sequencing of 120 samples to confirm presence of ORF8 deletions.

There are no line numbers in the draft so I have quoted the ms where relevant:

* The data presented in supplementary material show results across all SARS-CoV-2 genes, but in the main text and Figure 3, only Spike and ORF1a are shown as comparators for ORF8. I found it much more compelling to see the full complement of genes, and would recommend including this in the main text. This would also help the argument made in the Discussion that ORF7a/b may also be dispensable at least under some conditions.

* The authors briefly discuss a suggestion from literature that ORF8 mimics IL17, but at least one recent study (Matsuoka et al 2022, cited as #33 in this ms) found no evidence of cytokine-like effects, and indeed questioned the validity of the IL17 finding. The same study instead suggested that downregulation of MHC-I by secreted ORF8 might be advantageous for immune evasion within-host, while the intracellular ORF8 fraction might be disadvantageous because it induces ER stress. Although likely out of scope for the present study, it would be interesting to know whether the authors have attempted to look at their data within-host, eg by examining BAM files for evidence of sub-consensus nonsense mutations in ORF8 vs the observed high rates at the population level. At least one study I'm aware of (Li et al 2022, Cell Reports <https://doi.org/10.1016/j.celrep.2021.110205>) found that the overall mutation rate within-host was highest in ORF8.

* "This difference in cluster size could reflect different fitnesses associated with different types of gene knockout. For example, multiple non-singleton, large deletion clusters had deletions over 300bp, which resulted in knockout of both ORF8 and ORF7b" - this was somewhat confusing; I would suggest separating out deletions that span only ORF8.

* "This skewed distribution [of truncated protein size] alone suggests non-random process underlying premature stop codons." - I'm not sure I follow the logic. Any early stop will truncate the protein, and there are more ways to create a short protein than a long protein if multiple hits occur on a given gene (ie censoring - once truncated, any further mutations are irrelevant to protein size). What this observation suggests is that the overall rate of nonsense mutations in ORF8 is high.

* "The largest genes, ORF1a, ORF1b, and Spike, contained the most large deletions, with >24,000 in each compared to 1,517 large deletions in ORF8 (FigS3A)." - It would be helpful to normalize numbers per kilobase to make these more comparable.

* We used a χ^2 test to compare the number of individuals who were hospitalized or died due to SARS-CoV-2 infection by presence of an ORF8 knockout in their sequenced sample. - Could you comment on why a chi squared was used rather than Fisher's exact test?

* Figure 1 - please make clade labels and axis labels larger

* Maximum-likelihood phylogeny using IQ-TREE - please specify the selected model and settings

REVIEWER RESPONSES

Reviewer #1 (Remarks to the Author):

The authors investigate the repeated emergence of ORF8 knockout during SARS-CoV-2 evolution, which is primarily associated with Alpha and XBB variants (though across the other lineages/variants prevalence is ~3%). Given the frequency of ORF 8 knockouts, which now dominates current strains, the authors set out to test whether ORF 8 knockout is mildly deleterious or neutral and has hitchhiked with other adaptive mutations in the virus genome. They use two datasets, the global dataset collated to reconstruct the Usher phylogeny, and a local dataset from Washington state, where sequences with ORF 8 knockouts are sampled more evenly through time. To understand the role of selection in emergence and fixation of ORF8 knockouts, they undertook dN/dS analysis and examined the success of clades with ORF knockouts by calculating the cluster size. These findings were compared to key genes, which are necessary for viral replication, some of which have undergone adaptive evolution (e.g. spike).

Overall, I find the main conclusion that ORF8 knockouts are under positive selection and provide a fitness advantage to the virus compelling, but I wonder how robust these results are when the dataset is split by variants? Does the growth advantage and elevated nonsynonymous rates of ORF 8 knockouts hold across all variants, or just Alpha and XBB variants, where this mutation dominates? Are there differences in persistence times of the ORF 8 knockout clades between the two variants? Given that the virus would be associated with different population sizes (i.e. infections) and different levels of population immunity when Alpha and XBB variants emerged, I expect there would be some differences, which could provide further insight into the evolutionary impact of ORF 8 knockouts.

To address this question, we calculated dN/dS rates and the ratio of the geometric mean of nonsynonymous cluster size over the geometric mean of synonymous cluster size for each clade with greater than 500 samples. Due to the massive differences (over 1000x) in the ratio of largest nonsynonymous cluster over largest synonymous cluster associated with high dispersion of cluster size and the relatively few ORF8 mutations per clade, we were not able to estimate growth rate advantages per clade. Confidence intervals were wide for clades for these metrics, suggesting limited power to resolve if nonsynonymous mutations were increased or more successful on a per clade basis. However, 7 non-Alpha/XBB clades had significantly elevated nonsense dN/dS estimates, and 3 non-Alpha/XBB clades had significantly elevated nonsense cluster size ratios. This suggests that while there may be some heterogeneity in transmission advantage across clade; the signal is not only driven by Alpha & XBB. We have added these results to the main text (L315-348), methods (L663-678) and Supplementary Figure 9. Since ORF8 nonsense mutation was a clade-defining mutation for Alpha and an early mutation for XBB, we did not examine differences in persistence times between these clades as this comparison would essentially be only comparing persistence times between two clusters.

This analysis is included in the manuscript as:

“To further test if increased fitness for ORF8 knockout was driven by specific clades, we estimated ORF8 dN/dS rates split out by each Nextstrain clade, for each clade larger than 500 clusters (Fig S8A). Most clades had wide confidence intervals, with dN/dS rates for nonsense mutations indistinguishable from 1 for 17 clades. An additional eight clades – 19A, 20A, 20B, Alpha, Gamma, 21C, 21F, and 21H – had rates significantly greater than 1. Only four clades – 21K, 20G, 22D, 20F – had rates significantly less than one, which suggests that mutations are well-tolerated across the entire tree. A Wilcoxon Signed Rank test comparing if per-clade point estimates for dN were larger than dS was significant for missense mutations but not nonsense mutations ($p=0.0057$ & $p=0.24$ respectively).

Due to the massive differences (over 1000x) in the ratio of largest nonsynonymous cluster over largest synonymous cluster associated with high dispersion of cluster size and the relatively few mutations per clade, we were not able to reliably estimate growth rate advantages per clade. Instead, we compared the ratio of the geometric mean of nonsynonymous cluster size and geometric mean of synonymous cluster size for each clade (Fig S8B). Since each clade covers a smaller time window than the entire tree, the size bias from different cluster start times is minimized. Confidence intervals generated by bootstrapping across clusters were wide, suggesting we had limited power to determine if nonsynonymous clusters were larger than synonymous clusters. In 21J, 20A, Alpha, and 22C nonsense clusters were significantly larger on average than synonymous clusters; however, no clades had nonsense clusters significantly smaller on average than synonymous clusters. A one-sided Wilcoxon Signed Rank test comparing if per-clade point estimates for nonsynonymous cluster growth ratios were larger than synonymous cluster growth ratios was not significant for either missense mutations or nonsense mutations ($p=0.88$ & $p=0.39$ respectively).

Combining both dN/dS estimates and cluster size ratios for each clade, we see that 6 clades are differentiated from other clades by their elevated measures for nonsense mutations in both metrics, even if they failed to achieve significance levels (Fig S8C). These clades span a variety of time windows, which suggests that a time-correlated trend, such as development of high-levels of population immunity, does not drive the increased counts and success of ORF8 knockouts. In contrast, we identify only three clades with reduced dN/dS rates and smaller cluster sizes for nonsense mutations on average. Overall, these results suggest that the transmission advantage of ORF8 knockout may vary somewhat across clades, but Alpha and XBB alone are not the only clades with increased growth advantages.”

S8. ORF8 dN/dS and geometric mean cluster size ratios for missense and nonsense mutations by SARS-CoV-2 clades. For each Nextstrain clade with more than 500 samples in the SARS-CoV-2 UShER phylogeny, we estimated the following for ORF8: (A) dN/dS ratio and (B) the geometric mean size for nonsynonymous mutation clusters over geometric mean size for synonymous mutation clusters. Each estimate is split out by missense (teal) and nonsense (red) mutations. Panel (C) displays a scatterplot of the two estimates. Error bars represent 95% confidence intervals which were calculated by bootstrapping across nodes in each clade.

“Clade-level analysis

We labeled nodes in the UShER phylogeny with Nextstrain clades by passing clade labels from tip to parent nodes using a backwards traversal algorithm. If a node had multiple potential clade labels, the clade first identified was passed up to the parent node, i.e. 19A preceded 19B. We calculated ORF8 dN/dS for each clade with more than 500 samples using the same method applied to the entire phylogeny, except subsetting the tree to only nodes in that clade.

To test the growth advantage of nonsynonymous mutations over synonymous mutations for individual clades, we calculated the ratio of the geometric mean size of nonsynonymous clusters divided by the geometric mean size of synonymous clusters. We calculated the ratio split out by missense and nonsense mutations for each clade with more than 500 samples. Confidence intervals were generated by bootstrapping 10,000 times across nodes. We chose this approach, rather than modeling cluster growth rate, because it was more robust to biases from a single cluster, given the much smaller number of clusters in each clade relative to the entire tree. Since clades encompass smaller time windows than the entire phylogeny, average cluster size is less biased by different cluster starting times.”

I am less certain about their results indicating ORF 8 knockouts are associated with reduced severity since the odd ratios though < 1 are very close to 1. To me, this does not appear to be strong evidence for reduced severity, particularly to warrant screening for intact ORF8.

While odds ratios are very close to 1, we think the nearness of odds ratios reflects our power to identify an effect, not the strength of the effect itself. We have added power calculations to the manuscript demonstrating that we are only 22% powered to find a significant effect for hospitalizations & 75% powered to identify a significant effect for death (L375-377), methods (L709-715). Additionally, we have softened the language in the discussion around screening (L454-456).

“While the effect sizes estimated are barely significant, power analysis identifies only 22% power to identify a significant effect for hospitalization and 75% power to find an effect for death.”

“Currently, much of circulating SARS-CoV-2 has ORF8 knocked out; however, when ORF8 knockout arises on a viral backbone with intact ORF8 expression this suggests a transmission advantage. Conversely, rescue of ORF8 protein expression could increase the clinical severity of COVID-19 infections, though the effect may be small.”

“Specifically, we used the power test for the general linear model “f2.test” to estimate our power to identify the effect estimated by the general linear model. We calculated Cohen’s f^2 for ORF8 knockout as previously described⁷⁶ by the below equation:

$$f^2 = \frac{R_{AB}^2 - R_A^2}{1 - R_{AB}^2},$$

where R_{AB}^2 is the McFadden's R-Squared value for the model with all coefficients, including ORF8 knockout, and R_A^2 is the McFadden's R-Squared value for the model with all coefficients, except ORF8 knockout."

Furthermore, the discussion is quite limited, which mainly focuses on reiterating the key results. For example, I would like to see some discussion of why ORF 8 knockouts appear to dominate during Alpha and XBB variants, but not other lineages/variants? Why might ORF 8 knockout provide a transmission advantage in these two variants?

We have updated the discussion to discuss three different hypotheses for why ORF8 knockouts have not yet fixed globally and only dominated in Alpha and XBB (L479-494). Our hypotheses include: (1) viral backbone effects, which we find limited evidence for, (2) clonal interference, which is a strong candidate explanation, and (3) within-host vs. between-host fitness, which could contribute as ORF8 knockouts are not identified in chronic SARS-CoV-2 infections that are hypothesized to disproportionately contribute to SARS-CoV-2 evolution.

"Given the magnitude of the transmission advantage estimated for ORF8 knockout, it is puzzling that ORF8 knockouts have not been fixed and have only spread globally in Alpha and XBB subclades. While we identified some heterogeneity across clades in ORF8 dN/dS rates and nonsense-synonymous cluster size ratio, increased mutation counts and growth advantage were not limited to Alpha and XBB. Thus, although viral backbone could modulate fitness effects to some degree, it does not explain the few occurrences of ORF8 knockout reaching appreciable global frequencies. ORF8 knockout may be a classic example of clonal interference, especially considering the low probability of introducing a gene knockout and the high fitness boosts of mutations in other parts of the genome. For example, just as it appeared that ORF8 knockout from XBB descendants might globally fix, BA.2.86 viruses outcompeted XBB, dropping ORF8 knockout frequencies⁵⁷. The tug of war that Kim et al propose between within-host fitness and between-host fitness could also contribute to the lack of ORF8 knockout fixation. ORF8 nonsense mutations are absent from chronic infection associated mutations⁵⁸. Within chronic infections, intact ORF8 could provide a necessary edge to evade the host immune system. The hypothesized disproportionate contribution of chronic infections to global SARS-CoV-2 evolution could prevent ORF8 knockout fixation⁵⁹⁻⁶³."

Reviewer #2 (Remarks to the Author):

The study by Wagner and colleagues presents a succinct, well-described and compelling argument that global evolution of SARS-CoV-2 has included repeated loss of ORF8 function, and that ORF8 loss consistently confers evolutionary advantage. Although the frequent loss of ORF8 has been remarked upon by others, to my knowledge this is the first comprehensive

synthesis of the available evidence. This finding is supported by a rigorous analysis of relative growth rates of viral clades that have lost ORF8 expression either by nonsense mutation or via large deletions. The authors demonstrate that ORF8-knockout clades have a transmission advantage over other circulating viral clades, even when clades with the highest transmission advantage (such as Alpha) are excluded. A strength of the study is the inclusion of datasets at both the regional level (Washington State surveillance Feb 2020 - March 2023) and the global (Usher phylogeny). Also commendable are sensitivity analyses and validation work, including additional Sanger sequencing of 120 samples to confirm presence of ORF8 deletions.

There are no line numbers in the draft so I have quoted the ms where relevant:

* The data presented in supplementary material show results across all SARS-CoV-2 genes, but in the main text and Figure 3, only Spike and ORF1a are shown as comparators for ORF8. I found it much more compelling to see the full complement of genes, and would recommend including this in the main text. This would also help the argument made in the Discussion that ORF7a/b may also be dispensable at least under some conditions.

Thanks for the feedback! We have updated the main text figure (now Fig 4) to show results for all genes and added discussion of results for all genes to the main text (L245-255,260-261,290-293,296-299):

“In the structural (S, E, M, N) and replicase (ORF1a, ORF1b) genes, we identify strong selection against nonsense mutations, with dN/dS values <0.01 . This result is consistent with these genes being necessary for viral replication. The relative missense estimates for these genes are also largely consistent with expectation. For example, in spike, which has undergone substantial adaptive evolution^{14,18}, we observe relaxed selection against missense mutation compared to replicase genes ORF1a and ORF1b (with dN/dS values of 0.52, 0.38, and 0.32 respectively). In the accessory genes, which by definition are not necessary for viral replication, dN/dS values for both nonsense and missense mutations are elevated. ORF7a, ORF7b, and ORF8 all have especially high dN/dS values, with missense estimates >1 and nonsense estimates $>4.8x$ that of other genes. Uniquely, ORF8 has values >1 for both missense nonsense mutations (1.09 and 1.11 respectively).”

“Comparing across genes, we can conclude that negative selection on mutations in ORF8, ORF7a, and ORF7b are strongly weakened relative to other genes.”

“For all other genes, clusters with nonsense mutations grew at a reduced rate (0.07x on average) relative to clusters with synonymous mutations. These differences were not statistically significant, likely due to the very few number of nonsense mutation clusters observed in most genes resulting in wide CIs (Fig 4B). For example, in spike and ORF1a, 0.035% and 0.017% of mutations were nonsense respectively. Comparing cluster growth rates assumes mutations occurring and being sampled, so rates will be less sensitive for detecting clusters with mutations under strong negative selection. For

ORF7a and ORF7b, which had elevated counts of nonsense mutations, the absence of a difference in cluster growth rate between nonsense and synonymous mutations could be consistent with neutral evolution in these genes.”

* The authors briefly discuss a suggestion from literature that ORF8 mimics IL17, but at least one recent study (Matsuoka et al 2022, cited as #33 in this ms) found no evidence of cytokine-like effects, and indeed questioned the validity of the IL17 finding. The same study instead suggested that downregulation of MHC-I by secreted ORF8 might be advantageous for immune evasion within-host, while the intracellular ORF8 fraction might be disadvantageous because it induces ER stress. Although likely out of scope for the present study, it would be interesting to know whether the authors have attempted to look at their data within-host, eg by examining BAM files for evidence of sub-consensus nonsense mutations in ORF8 vs the observed high rates at the population level. At least one study I'm aware of (Li et al 2022, Cell Reports <https://doi.org/10.1016/j.celrep.2021.110205>) found that the overall mutation rate within-host was highest in ORF8.

We examined the rate and frequency nonsense mutations in intrahost single nucleotide variants from four sequencing runs (1300 samples) at the University of Washington from August & Sept 2022. We identified an elevated within-host rate and within-host frequency of nonsense mutations in both ORF7b and ORF8, relative to other SARS-CoV-2 genes, suggesting relaxed within-host purifying selection for gene knockout. This is consistent with identified positive selection for ORF8 knockout at a population-level, and the high dN/dS rates for ORF7b at a population level. We have added these results to the main text (new Figure 3) (L194-217) as well as methods (L584-601):

“Deleterious mutations are often under purifying selection within a host. If the high rate of ORF8 knockout observed in consensus sequences extended to within-host frequencies, this result would additionally argue against deleterious fitness associated with ORF8 knockout. Therefore, we examined the rate of nonsense mutations in intra-host variants in a subset of 1,015 SARS-CoV-2 samples that did not have a consensus-level stop codon, which were sequenced from August-September 2022 in WA (Fig 3). We defined nonsense intrahost variants as single nucleotide polymorphisms creating a premature stop codon and were present in 1-50% of reads covering that site. Intrahost nonsense variants had to be further supported by at least 10 reads, with a total read of coverage of at least 100 for the site. ORF7b had the highest per codon rate of intrahost nonsense mutations (8.9×10^{-4}) followed by ORF8 (2.4×10^{-4}). Both genes had elevated intrahost frequencies of nonsense mutations relative to all other genes (ORF7b median = 0.036, ORF8 median = 0.032, other genes median = 0.015) (Fig 3A). Differences in allele frequency between nonsense mutations in ORF7b/ORF8 and other genes were statistically significant (ORF7b: $p = 5.3 \times 10^{-11}$, ORF8: $p = 4.1 \times 10^{-8}$, Wilcoxon Rank Sum Test) (Fig 3B). These results are consistent with increased population-level ORF8 knockout and suggest altered within-host selection pressures on both ORF8 and ORF7b.

Fig 3. Intra-host nonsense mutations in ORF7b and ORF8 occur at higher rates per codon and at higher allele frequencies compared to other genes. We tested the intra-host variants of 1,015, high-coverage SARS-CoV-2 samples sequenced in Washington State from August to September 2022, which did not have a consensus level premature stop codon for nonsense mutations. (A) shows the per codon, per sample rate of intra-host nonsense mutations in each gene. The frequencies of nonsense mutations observed in intra-host variants are shown in (B) for each gene. Black lines indicate the median frequency.”

“Intrahost analysis

We examined the rate and frequency of nonsense mutations in intrahost single nucleotide variants in 1,300 SARS-CoV-2 samples sequenced by the University of Washington from August to September 2022 as part of a genomic surveillance program. Nasopharyngeal, nasal, or oropharyngeal swabs with PCR cycle threshold < 31 were randomly selected and sequenced as described previously⁷³. Briefly, after RNA extraction, library preparation was performed using the Illumina COVIDseq protocol with ARTIC v4.1 primers (Integrated DNA Technologies). Prepared libraries were pooled and sequenced on an Illumina Novaseq6000 instrument using a 2x150 read format targeting at least 1 million reads per sample. Genome assembly was performed using a custom pipeline (https://github.com/greninger-lab/covid_swift_pipeline) which performs trimming to remove adapters and low quality regions, primer clipping, variant calling, and consensus genome generation. We excluded 55 samples due to inadequate coverage (>10% N’s or <7,419 reads, which was two standard deviations under the mean coverage) or poor amplification (>25% of reads trimmed). We excluded an additional 230 samples with a consensus-level premature stop in any gene to avoid biasing rates of intrahost nonsense mutations. In the remaining 1,015 samples, we required all intrahost nonsense variants to have $\geq 1\%$ frequency with a variant coverage of 10x and a total position coverage of 100x.”

* “This difference in cluster size could reflect different fitnesses associated with different types of gene knockout. For example, multiple non-singleton, large deletion clusters had deletions over

300bp, which resulted in knockout of both ORF8 and ORF7b” - this was somewhat confusing; I would suggest separating out deletions that span only ORF8.

Thanks for the suggestion; we have updated the text to include the comparison of cluster size for deletions that only knockout ORF8 as well as those affecting ORF7b (L129-136).

* “This skewed distribution [of truncated protein size] alone suggests non-random process underlying premature stop codons.” - I’m not sure I follow the logic. Any early stop will truncate the protein, and there are more ways to create a short protein than a long protein if multiple hits occur on a given gene (ie censoring - once truncated, any further mutations are irrelevant to protein size). What this observation suggests is that the overall rate of nonsense mutations in ORF8 is high.

In the WA phylogeny, most clusters are not nested, i.e. only one stop mutation has occurred along the branches leading to that tip, so I think this is unlikely to be an example of censoring. However, opportunities to add a stop mutation are limited given the sequence composition, so it’s likely inappropriate to read too much into the skewed distribution. We have updated the main text accordingly (L151-153)

* “The largest genes, ORF1a, ORF1b, and Spike, contained the most large deletions, with >24,000 in each compared to 1,517 large deletions in ORF8 (FigS3A).” - It would be helpful to normalize numbers per kilobase to make these more comparable.

In Fig S3B, we plot the number of large deletions per kb per sample to better compare cross genes. We have added to the text here to more clearly highlight this comparison (L163-167).

* We used a χ^2 test to compare the number of individuals who were hospitalized or died due to SARS-CoV-2 infection by presence of an ORF8 knockout in their sequenced sample. - Could you comment on why a chi squared was used rather than Fisher’s exact test?

We used a χ^2 because we deemed the sample sizes (at minimum, 383 and 129 for hospitalization and died tests respectively) large enough for the approximation to not impact our interpretation of the test and to avoid any computation issues. A Fisher’s exact test identifies similar significant differences in the proportion of people who died due to a SARS-CoV-2 infection (p-value: 8.21e-05) or were hospitalized due to SARS-CoV-2 infection (p-value: 3.08e-05) and the presence of an ORF8 knockout in their sequenced virus sample. We have updated the main text and methods with Fisher’s exact test results since a χ^2 is unnecessary in this case given only a two-way contingency table (L359,361-362,688).

* Figure 1 - please make clade labels and axis labels larger
Done. We’ve increased the axis & clade labels on the tree in Figure 1.

* Maximum-likelihood phylogeny using IQ-TREE - please specify the selected model and settings

We have added the following text into the methods: “For IQ-TREE, we specified a GTR substitution model, 10 initial parsimony trees, and four unsuccessful iterations to stop; for TreeTime, we used a substitution rate of 0.008 with a standard deviation of 0.004.” (L541-543)

Reviewer #1 (Remarks to the Author):

All my previous concerns have been addressed and have no further comments. Many thanks to the authors for their efforts in robustly revising the manuscript, I think it further strengthens their findings and will be of interest to a broad research community.

Reviewer #2 (Remarks to the Author):

The authors have amended the manuscript appropriately and my comments have been fully addressed. The new intrahost analysis is a particularly strong addition; it shows reduced purifying selection in ORF8 and (even more strongly) ORF7b, consistent with the observed repeated emergence of knockouts at population level.

Reviewer #2 (Remarks on code availability):

I have examined the repo and the notebooks; the code is present and appears correct, but I have not attempted to install and run it. A minor note that Fig2 is missing ORF1ab in the example dataset and so the code in the notebook doesn't show one of the plots.

REVIEWER COMMENTS

Reviewer #1:

All my previous concerns have been addressed and have no further comments. Many thanks to the authors for their efforts in robustly revising the manuscript, I think it further strengthens their findings and will be of interest to a broad research community.

Reviewer #2:

The authors have amended the manuscript appropriately and my comments have been fully addressed. The new intrahost analysis is a particularly strong addition; it shows reduced purifying selection in ORF8 and (even more strongly) ORF7b, consistent with the observed repeated emergence of knockouts at population level.

I have examined the repo and the notebooks; the code is present and appears correct, but I have not attempted to install and run it. A minor note that Fig2 is missing ORF1ab in the example dataset and so the code in the notebook doesn't show one of the plots.

We thank the reviewers for their feedback on the manuscript; their edits and suggestions have improved the quality of the manuscript and robustness of the results. We have updated the code in the Fig2 notebook, so that all plots are displayed.